# Predictors of Eating Less Meat and More Plant-Based Food in the Polish Sample

**DOI:** 10.3390/nu16111646

**Published:** 2024-05-27

**Authors:** Marzena Jeżewska-Zychowicz, Marta Sajdakowska, Jerzy Gębski, Małgorzata Kosicka-Gębska, Krystyna Gutkowska

**Affiliations:** Institute of Human Nutrition Sciences, Warsaw University of Life Sciences (SGGW-WULS), Nowoursynowska 159 C, 02-776 Warsaw, Poland; marzena_jezewska_zychowicz@sggw.edu.pl (M.J.-Z.); jerzy_gebski@sggw.edu.pl (J.G.); malgorzata_kosicka_gebska@sggw.edu.pl (M.K.-G.); krystyna_gutkowska@sggw.edu.pl (K.G.)

**Keywords:** food intake, adults, a cross-sectional study

## Abstract

The study aimed to identify predictors of the intention to eat less meat and more plant-based foods, including attitudes towards eating meat, habitual meat eating, subjective norms, and self-identity. A cross-sectional study using CAWI (Computer-Assisted Web Interview) was conducted in a group of 1003 Polish adults in 2023. To measure the predictor variables, the following tools were used: Beliefs and Eating Habits Questionnaire (KomPAN), Meat Attachment Questionnaire (MAQ), and scales to measure subjective norms and self-identity. Logistic regression analysis was used to verify associations between independent variables, and the intentions to eat more plant-based food and less meat next year were treated as dependent variables. More respondents were willing to increase their consumption of plant-based foods rather than reduce their meat consumption. The intention to consume less meat and more plant foods was more prevalent among women, older people (only intention to reduce meat consumption), and better-educated people (only intention to increase plant food consumption). Habitual frequency of eating plant foods, negative feelings about meat, and environmentally oriented identities had a stimulating effect on the intention to eat more plant foods and less meat, while experiencing pleasure in eating meat had a limiting effect on the intention to eat more plant foods and less meat. In addition, the habitual frequency of meat consumption and subjective norms reduced the likelihood of eating less meat, while no predictive effect was observed for the intention to eat more plant foods. In conclusion, educational and promotional activities to raise awareness of the link between food consumption and the environment can have a strong impact on eating less meat and more plant-based food, even among those strongly accustomed to meat consumption.

## 1. Introduction

Dietary guidelines recommend reducing meat intake, pointing in particular to the need to limit red and processed meat intake [1], and advocate for a plant-based diet with whole grains, vegetables, fruit, legumes, nuts, and unsaturated oils [2]. However, a shift from animal-based to plant-based foods is a challenge for humans for several reasons, mainly related to the special place of meat in the culture of many countries, including Poland [3,4,5].

Previous studies indicate that reducing meat intake is strongly associated with motives related to health [6,7] and sustainability, including respect for animal welfare [8,9]. Health consciousness and perceptions of the environmental impact of meat substitutes are also predictors of meat substitute consumption [10,11,12]. Moreover, recent findings indicated that consumers oriented towards health and naturalness were more willing than others to reduce their meat consumption and adopt a plant-based diet. In contrast, attachment to cultural traditions, especially in Western countries, makes it difficult to reduce meat consumption [3,4,13]. Previous research on the declarative importance of health, environmental, and traditional motives in determining meat consumption confirms their importance but also highlights differences stemming from the cultural background. However, few studies have considered identity concerning health, the environment, and tradition and have not related it to meat choice [14].

The importance of health, sustainability, and tradition can differentiate beliefs about food, including meat and plant-based food. In line with some previous research, a plant-based diet is seen as inconvenient because of the difficulty of preparing meals, as it requires a lot of time, skill, expensive ingredients, etc. [12,15,16,17,18,19]. Moreover, according to another study, it is not easy to find available options for meatless meals [20]. Some studies comparing the acceptance of both types of food show that plant-based food is less preferred than food of animal origin due to certain sensory characteristics, including taste [21,22,23,24]. Perceptions of plant-based foods may reflect the social context, including stereotypes associated with such foods, e.g., the belief that a diet based on plant-based foods reduces physical performance because it is deficient in essential components necessary for the body to function normally, which is linked to the belief that such a diet is unsuitable for men [25].

Meat is firmly rooted in human food traditions [3,4,26]. Tradition favours its consumption on festive occasions, but nowadays meat consumption has also become a highly habitual behaviour [27]. Thus, meat plays an important role in the diet of many people [28]. For a substantial number of consumers, meat is synonymous with wealth and a wholesome diet [29]. Except for some religions, meat consumption is perceived as a normal and evolutionarily natural [29]. In Poland, meat consumption is also strongly rooted in tradition [5], and the preparation of meat meals is perceived as convenient, similarly to other countries [30,31]. In the past, meat consumption was associated with typical masculine attributes [32,33,34]. Today, the link between masculinity and meat consumption is increasingly being questioned due to alternative images of masculinity [35]. According to Rees and colleagues [27], meat consumption is largely driven by non-reflective and automatic processes influenced by situational contexts, limiting the importance of conscious cognitive processes when making consumption decisions. Beliefs about meat based on knowledge of its health benefits and one’s own experiences may determine its consumption in different ways. Indeed, research shows that some people believe that meat consumption is unhealthy [18], but others see meat as healthy [36]. There is a strong relationship between meat consumption, the perception of taste [31], and the expectation of pleasure after eating it [29]. People who are strongly attached to meat consumption are less likely to change their dietary habits [29,37,38]. Indeed, habits and routines in eating meat constitute some of the main barriers to reducing meat consumption [39]. The fact that meat is perceived as more enjoyable than plant-based food [8,40] makes it difficult to shift from a meat-based diet to a more plant-based one.

Reluctance to change, and the difficulties associated with it, can be linked to the expectations of others, as eating is a social situation. Many studies to date have shown that subjective norms significantly condition eating behaviour [41,42]. Normative eating behaviours, such as the perceived appropriateness of food in a situational context, appear important for food acceptance [43,44] and affect food choices [45]. Thus, the widespread belief that eating meat is a social norm and is evolutionarily natural [29] will encourage meat eating rather than reducing its consumption. Thus, if a person knows what other people think, feel, or do when it comes to eating meat, then when a person is observed or is aware of normative expectations, they will adjust their attitude and behavior to conform to the existing social norm [46]. The importance of subjective norms in conditioning people to eat meat can be further reinforced by existing stereotypes, as they can be used to create a certain impression on others [25]. For example, a study in Italy found that women prefer omnivorous men and perceive them as more attractive than men who follow a plant-based diet [47]. However, the results of studies on the impact of subjective norms on consumers’ intention to eat meat and plant-based food are inconclusive. Some studies report that there is no such relationship [48], while others confirm that such a link exists [14,49,50]. Since Polish cuisine is traditionally heavily based on meat, vegans and vegetarians often experience a lack of acceptance of their diet from family and friends, and they face unpleasant situations because of the diet [25]. Further research should, therefore, include subjective norms alongside other variables to better understand their importance in determining the eating of animal and plant-based foods, but also the transition to a more plant-based diet.

According to the theory of planned behavior (TPB), attitudes toward behavior and subjective norms are factors that explain behavioral intention well [51], which has also been confirmed for meat-eating [52]. It has been suggested that eating habits are a construct that should be added to the TPB to predict food choices and dietary behaviors [53,54] because it is a good predictor of intention to eat certain types of food (i.e., eggs, fish). Nonetheless, the importance of habitual eating in predicting intentions to eat meat cannot be considered unequivocal [42,52]. At the same time, eating habits can weaken the effects of both attitudes and subjective norms [54], prompting the simultaneous inclusion of these factors in studies. Identity as a behavioral motivator is also linked to meat consumption [55]. All of these factors have a proven role in conditioning meat consumption, although they have rarely been considered simultaneously in studies. In addition, the importance of intentions to eat more plant-based foods has not been studied. Instead, it can be assumed that both attitudes toward meat eating, habitual meat eating, and subjective norms regarding meat eating may also determine a relationship with intentions to eat more plant-based foods. Thus, the study aimed to identify predictors of intention to eat less meat and more plant-based foods by looking into attitudes towards eating meat, habitual meat eating, subjective norms, and self-identity. The findings of this study contribute to and extend the understanding of the consumers’ intentions to change meat and plant food consumption behaviors.

## 2. Materials and Methods

### 2.1. Study Design and Sample Collection

A cross-sectional survey was conducted between June and September 2023 following the ESOMAR (European Society for Opinion and Marketing Research) code of conduct using the CAWI (Computer-Assisted Web Interview) technique. Data confidentiality, as well as anonymity, was assured. Moreover, the study was approved by the Ethics Committee of the Warsaw University of Life Sciences, in Poland (Resolution No. 8/RKE//2023/U, 20 April 2023). It was conducted in agreement with the guidelines of the Declaration of Helsinki. The following inclusion criteria were taken into account: gender (women and men), age (18 years and older), eating meat at least once a week, and providing informed consent to participate. The exclusion criteria were: age under 18, eating meat less than once a week, and lack of informed consent to participate. The study sample included 1003 Polish adults from all regions of the country. The study sample was recruited by a professional research agency. The selection criteria of the sample considered the representativeness of the Polish population due to the province and the quota character by gender, education, and place of residence.

### 2.2. Dietary Data

The frequency of consumption of the selected food groups was assessed with the Beliefs and Eating Habits Questionnaire (KomPAN) [56], which was validated in Polish adults [57]. The participants reported the habitual frequency of eating six groups of food: cold cuts and sausages, red meat (pork, beef, veal, lamb, game), white meat (chicken, turkey, rabbit), legumes (beans, peas, soybeans, lentils), vegetables, and fruit in the 3 months preceding the survey using one of the answers: 1—less than once a month or never; 2—1–3 times a month; 3—once a week; 4—a few times a week; 5—once a day; and 6—a few times a day. During the data analysis, the answers were converted to reflect the daily frequency of intake, ranging from 0—less than once a month or never; 0.06—1–3 times a month; 0.14—once a week; 0.5—a few times a week; 1—once a day; and 2—a few times a day [58]. The habitual intake of food products was calculated for plant- and animal-based food separately by summing the daily frequencies of intake of groups belonging to those categories.

Besides the questions on consumption frequency, respondents declared the intention to eat more plant-based food and less meat next year. The questions were as follows: “Do you intend to eat more plant-based food next year?” (yes/no) and “Do you intend to eat less meat next year?” (yes/no).

### 2.3. Other Variables

#### 2.3.1. Subjective Norm

Normative beliefs about eating less meat were measured for one group: namely “the significant others”. Respondents indicated whether their “significant others” think that they should eat less meat. Scores were rated on a five-point scale ranging from 1 (totally disagree) to 5 (totally agree). Respondents were then asked to what extent they were motivated to comply with the expectations of “the significant others”. Scores were rated on a five-point scale ranging from 1 (totally not) to 5 (very much). Subjective norms were computed by summing the scores of the normative belief and the corresponding motivation to comply. Scores for the subjective norms were in the range of 2 to 10.

#### 2.3.2. Attitudes towards Meat Consumption

The attitude towards eating meat was measured using seven statements from the Meat Attachment Questionnaire (MAQ) developed by [37] and two statements relating to the fondness of eating meat linked to its cultural meaning [59]. Based on these statements three scales were computed to measure the attitude toward meat:

Hedonism scale (A: Hedonism). The scale includes three statements: To eat meat is one of the good pleasures in life; I love meals with meat; A good steak is without comparison. Scores were rated on a five-point scale ranging from 1 (totally disagree) to 5 (totally agree). Cronbach’s alpha of this scale was 0.87. The mean score for hedonic attitude was computed. The higher the score, the greater the pleasure of eating meat.Negative feelings scale (A: Negative feelings). The scale includes three statements: Eating meat is disrespectful towards life and the environment; Meat reminds me of diseases; and By eating meat I’m reminded of the death and suffering of animals. Scores were rated on a five-point scale ranging from 1 (totally disagree) to 5 (totally agree). Cronbach’s alpha of this scale was 0.81. The higher the score, the more negative emotions accompany meat consumption.Beliefs about eating meat (A: Beliefs about eating meat). The scale includes three statements: Eating meat is a natural and indisputable practice; It is hard to imagine any celebration without meat dishes; Meat is an integral part of the diet of Poles resulting from tradition. Scores were rated on a five-point scale ranging from 1 (totally disagree) to 5 (totally agree). Cronbach’s alpha of this scale was 0.77. The higher the score, the more meat consumption is considered to be a part of the culture.

#### 2.3.3. Self-Identity

In line with both self-labeling self-identity theory and the symbolic interactionism approach, we used self-descriptors to measure self-identity [60]. Respondents rated themselves in terms of the importance they attribute to three values, namely health, environment, and tradition, by responding to two statements: I consider myself to be: a person who cares about health (health identity); a person who values tradition (traditional identity); a person who is oriented towards the environment and its protection (environmental identity). Scores were rated on a five-point scale ranging from 1 (totally disagree) to 5 (totally agree).

#### 2.3.4. Sociodemographic Variables

Questions on socio-demographic characteristics considered: age (in years), place of residence, education, and self-reported financial situation.

### 2.4. Statistical Analysis

Descriptive statistics were used to present the sociodemographic characteristics of the study sample. Data were presented as a sample percentage (%) for categorical data or mean and standard deviation (SD) for continuous data. The normality of the distribution of continuous variables was assessed with the normality Kolmogorov–Smirnov test, Lilliefors test, and normal probability plot. The independence Chi-square test and the Student’s *t*-test were used to determine if there were differences between subgroups. The level of statistical significance was set to *p* < 0.05. The strength and direction between the two variables were measured with Pearson correlation coefficient (r).

Logistic regression analysis was used to verify associations between habits (frequency of eating meat-based food, frequency of eating plant-based food), subjective norms, attitude towards meat (hedonic score, negative feelings, beliefs about eating meat), self-identity (independent variables), and the intentions to eat more plant-based food and less meat next year (dependent variables). Odds ratios (OR) represented the probability of belonging to a group declaring a change in consumption, i.e., eating more plant-based food (Model 1) and less meat and meat products next year (Model 2). The reference groups (OR = 1.00) were those who did not declare any change. Wald’s test was used to assess the significance of ORs.

The statistical analysis was carried out using IBM SPSS Statistics for Windows, version 29.0 (IBM Corp, Armonk, NY, USA).

## 3. Results

### 3.1. Description of the Study Sample

Table 1 presents the sociodemographic characteristics of the study sample. The sample consisted of 1003 adults, with 51.8% of women. The participants were aged 18–83 years; the mean age was 45.4 years (standard deviation—15.5).

### 3.2. Intention to Eat More Plant Food and Less Meat Next Year

More respondents reported plans to increase their consumption of plant-based foods (60.4%) than to decrease their consumption of meat (42.1%) in the following year. More women than men declared an intention to increase their consumption of plant-based foods (67.7% and 52.6%, respectively), but also an intention to decrease their meat consumption (48.3% and 35.4%, respectively). The number of people declaring the intention to reduce their meat consumption increased with age. In contrast, place of residence and declared financial situation did not differentiate either the intention to increase plant food consumption or to decrease meat consumption (Table 1).

### 3.3. Eating Habits, Subjective Norms, Attitudes towards Meat, Self-Identity, and Intentions to Eat More Plant Food and Less Meat in the Study Sample

The characteristics of variables considered in explaining the intention to eat less meat and more plant foods are presented in Table 2.

Respondents who declared an intention to eat more plant-based foods in the following year were characterized by a higher habitual frequency of eating plant-based foods (fruits, vegetables, legumes) and more negative feelings accompanying meat consumption. In addition, these individuals perceived themselves as more health-conscious and environmentally oriented compared to those who did not make such declarations. At the same time, the former group of respondents was characterized by a lower intensity of hedonic attitudes towards meat and by views indicating lower importance of tradition in conditioning meat consumption. The habitual frequency of eating meat products, subjective norms, and self-perception as a person who values tradition did not differentiate the declarations related to the consumption of plant-based food in the following year (Table 3).

On the other hand, respondents who intended to consume less meat products were characterized by a higher frequency of eating plant foods and a lower frequency of eating meat. In addition, these individuals exhibited a higher intensity of negative emotions associated with eating meat and perceived themselves as focused on the natural environment. Respondents who did not declare plans to reduce meat consumption, on the other hand, were characterized by higher scores for subjective norms, hedonic attitudes toward meat, and views indicating the importance of tradition in conditioning meat consumption, as well as a higher score for traditional identity. Health identity did not differentiate respondents’ intentions to eat less meat in the following year (Table 3).

### 3.4. Associations between Intentions to Eat More Plant Food and Less Meat and Eating Habits, Subjective Norms, Attitudes towards Meat, and Self-Identity

Associations between eating habits, subjective norms, attitudes towards meat, self-identity, frequency of eating plant food and meat food, and intentions to eat less meat and more plant food are presented in Table 4. The strongest associations (r > 0.4) were found between intentions to eat less meat and the score for hedonic attitude (negative correlation), as well as intentions to eat less meat and the score for negative feelings about eating meat (positive correlation). The latter correlated most strongly positively with intentions to eat more plant-based foods (r = 0.327). A positive relationship was found between the intention to eat more plant foods, as well as less meat and three variables: negative feelings (0.327 and 0.415, respectively), environmental identity (0.224 and 0.158, respectively), and habitual frequency of eating plant food (0.221 and 0.140, respectively). In contrast, beliefs about eating meat being a tradition correlated negatively with both the intention to eat more plant-based and the intention to eat less meat (Table 4).

A weak positive relationship was found between self-perception as a health-conscious, tradition-valuing, and environmentally oriented person and frequency of plant-based food consumption. More frequent consumption of meat products correlated positively with scores for subjective norms, hedonic attitudes toward meat, beliefs about eating meat, and traditional identity (Table 4).

### 3.5. Predictors of Eating More Plant Food and Less Meat in the Study Sample

Logistic regression results showing the predictive effect of the variables included in the study are presented in Table 5. The habitual frequency of plant food consumption and the environmental identity increased the chances of consuming more plant food in the following year by about 50%. In contrast, a more positive hedonic attitude toward meat reduced the chance of consuming more plant-based foods by about 40%.

The likelihood of reducing meat consumption increased with an increase in the habitual frequency of eating plant-based foods (by 31%), perceiving oneself as an environmentally oriented person (by 28%). Most importantly, a 2-fold increase in the likelihood of such declarations was observed when negative emotions associated with eating meat increased.

Being less likely to reduce the amount of meat consumed was associated with eating meat more often (by 21%), considering the subjective norm when eating meat (by 13%), and, most importantly, representing a hedonistic attitude toward meat (by 54%) (Table 5).

## 4. Discussion

The results of the study show that more people are willing to increase their consumption of plant-based foods than reduce their meat consumption. Similarly, previous studies have indicated that the willingness of many Western consumers to replace meat with meat substitutes is quite low [61], and this is especially observed among regular meat eaters. As an example, in an Australian study [17] and in a Finnish study, 46% and 48% of respondents, respectively, [62] who reported consuming beef regularly did not plan any changes. The reluctance to change may stem from the fact that many people feel psychologically attached to meat [37] and perceive meat consumption as too socially normal, evolutionarily natural, and hedonically pleasurable to stop eating it [29].

### 4.1. Sociodemographic Characteristics vs. Intention to Eat More Plant Foods and Less Meat

As in previous studies [11,17,63], it was confirmed that more women than men intend to increase their intake of plant foods and decrease their meat consumption. The fact that meat is associated with typical masculine attributes [32,33,34] may explain to some extent men’s reluctance to reduce meat consumption. Women, on the other hand, are more likely to adopt meat-reducing strategies, e.g., meatless meals [64,65]. Moreover, women also are more likely to be frequent eaters of pulses and plant proteins than men [10,66]. More frequent consumption of plant foods in women, as well as their more positive attitudes towards a pro-environmental protein intake [67] may prompt them to continue to increase their intake of plant foods.

Previous studies have found that age has an inconsistent association with a willingness to change one’s diet to one that is richer in plant foods and poorer in meat [68,69,70,71]. The study by Neff [70] showed that older consumers were more likely to reduce or consider reducing their consumption of some types of meat, such as red meat, which was also confirmed by the results of our study. Since older people are described as more traditional [72], and meat in many cultures, including Poland [3,4,5,73], is firmly embedded in culinary traditions, some interpretive inconsistency arises. In explaining these results, it is important to keep in mind that the complexity of different motives determines decisions about food choices, including meat [74]. Findings of the study by Hielkema and Lund [68] have shown that older people were less likely to decrease their meat consumption. In addition, it has been shown that older consumers have more positive beliefs about red meat than younger consumers and are more likely to be beef lovers [69]. Interestingly, the differences shown in intentions to reduce meat consumption after accounting for age were not observed in intentions to eat more plant foods. Also, younger respondents did not report more willingness to eat plant-based foods, even though younger consumers are more likely to adopt a vegetarian diet than middle-aged or older consumers [71].

Better educated people, on the other hand, are more likely to increase their consumption of plant-based foods, as shown in our research and other studies [75,76]. It was also found that consumers with higher education were more likely to be vegetarians than consumers with lower education, and they were more likely to choose plant protein alternatives to reduce their meat consumption [75,76]. However, our study did not confirm an association between education and the intention to eat less meat. Previous studies conducted in Poland have also shown that there is no relationship between education and the consumption of red meat, poultry, and cold cuts [77], but also that the relationship varies between men and women [73]. Unlike eating less meat, a positive relationship was shown between education and intention to eat more plant-based foods. This may be because more educated individuals are more exposed than others to novelties and varieties of food [78], but also because of their pro-environmental beliefs [79].

### 4.2. Predictors of Eating More Plant-Based Foods

It is not surprising that the affective component of attitudes toward meat, represented in the study by negative feelings and hedonism related to meat, was found to be significant in conditioning the consumption of plant foods. Taking pleasure in eating meat promotes greater meat consumption, which at the same time can be an important barrier to increasing the consumption of plant foods, such as those rich in protein [29,36], as confirmed in our study. Plant-based foods are commonly perceived as less tasty than meat [40,80], hence its consumption is probably not motivated by the pleasure of consumption but by other factors. For example, Graça [12] showed that pleasure orientation is less important, while naturalness orientation is more important for consumers following a plant-based diet than for other groups. Negative feelings associated with meat favor limiting its consumption, while at the same time, they may favor the consumption of other protein-rich foods [36], as confirmed by our results. The affective component of attitude is strongly linked to food preferences [81]. Plant-based foods are less preferred due to certain sensory attributes [21,23,24], which is not conducive to their consumption. The way to enhance preference is to get familiar with the food and, above all, to taste and consume it [82], as evidenced by the association between more frequent consumption of plant foods and declarations of increased consumption in the future. Preferences for plant foods have not been studied, so including them in future research would provide more comprehensive insights into the determinants of plant food consumption.

People focused on environmental issues were more likely to eat more plant-based foods in the future, which is consistent with the results of previous studies [12,83,84,85]. This orientation is conducive to reducing meat consumption [11,86], and also abandoning the consumption of meat and other animal products altogether [87]. In addition, people with pro-environmental attitudes are more likely to accept new forms of food if they are convinced of their positive environmental impact [88]. Nevertheless, even if the pro-environmental motive is a crucial one for formulating behavioral change intentions, it is not sufficient to lead toward a behavioral change [89]. Plant-based diets are generally considered acceptable in low- and low-middle-income countries, while in high-income countries people often consider meat an important part of a meal. It is so because cultural, financial, ethical, and religious reasons can influence food choices more than pro-environmental attitudes. Thus, to make plant-based diets more socially desirable, it is necessary not only to focus on the environmental aspect but also to promote the broader context of plant-based foods, i.e., the links between health, naturalness, sustainability, and these foods [90]. Although the results of our study did not show a predictive effect of health orientation on the intention to eat more plant foods, such a holistic approach may increase the chances of altering the ratio between plant and animal foods in the diet. Nevertheless, previous studies have also shown that one can expect variation in the predictive effect from different factors within such a general construct as plant food.

In the remote past, plant food dominated the diet of Poles, while the consumption of meat signified membership in the upper social class, and thus it gained high social importance and was desirable. The tradition of eating meat as a food carrying social meaning could explain the negative relationship between intentions to eat more plant-based foods and attachment to tradition. It seems that dietary recommendations and awareness of the negative consequences of consuming large amounts of meat [91] may contribute to changing the social status of meat and recognizing the great importance of plant foods in our ancestors’ diets.

Previous research suggests that individuals who are strongly attached to meat consumption are less likely to change their dietary habits [37], which may explain the lack of association between the frequency of eating meat and intentions to eat more plant-based foods, even though one would expect a negative effect. In addition, beliefs about eating meat, subjective norms, and the aforementioned health identity were not predictors of intentions to eat plant foods. The lack of a link between caring about health and the desire to eat more plant-based foods shows that the positive health effects of plant-based foods presented by scientists [92] do not suffice to convince consumers to eat more of such foods. Identifying effective methods to improve nutritional knowledge is a challenge [93]. However, practical information about the preparation of tasty plant-based meals should also be of great importance [94]. For more people to adopt a plant-based diet, there is, therefore, a need to improve their food knowledge, as well as their skills to cook and organize tasty meals [89]. Both the lack of knowledge and the lack of skills can discourage the introduction of plant-based products, especially unfamiliar ones. It could be expected that the opinions and expectations of one’s significant other to reduce the amount of meat consumed could increase interest in the consumption of plant-based foods. However, to date, the research results in this area are ambiguous. Some of them, including our study, report the lack of influence of subjective norms on consumers’ intention to eat plant food [48], while others confirm the existence of such a relationship [49,50].

### 4.3. Predictors of Eating Less Meat

The results obtained, as well as those of other studies [29,32,37,64], indicate the importance of attitudes towards meat eating in both determining actual meat consumption and the decision to reduce it in the future. It has been confirmed that meat consumption shows a strong association with the expectation of pleasure from its consumption [29], and this in turn is a major barrier to reducing meat consumption and/or adopting a vegetarian diet [36]. Experiencing negative emotions associated with eating meat, on the other hand, has the opposite effect. Namely, it favors the reduction of meat consumption. Disgust as a negative emotion accompanying food is indicated as one of the most commonly reported reasons for vegetarianism and reduced meat consumption in Western countries [36].

Although more people identifying themselves as traditionally and environmentally oriented declared an intention to eat less meat, only for the environmental identity was a predictive positive effect found. This confirms that environmental concerns are one of the key reasons for vegetarianism and limited meat consumption [36]. Yet, consumers’ awareness of the environmental impact of meat consumption is still low [61]. In addition, even if sustainability awareness is identified as a key element, it is not sufficient to bring about behavioral change [89]. Nevertheless, as a result of such research, the predictive effect of different motives can be compared at the level of intention and behavior, and the question can be answered as to which motive of the stated intention is most conducive to the occurrence of behavior [29].

The relationship between the frequency of meat consumption and self-perception as a person who values tradition suggests that the tradition of eating meat, often identified as a barrier to reducing meat consumption, continues to show a link with current behavior. However, its importance may change in the future. This may be evidenced by the lack of a predictive effect instead of the expected negative correlation with intentions to reduce meat consumption. Like traditional identity, self-perception as a health-conscious person was also not found to be related to the intention to reduce the amount of meat consumed. By contrast, concerns about one’s health are among the most frequently cited reasons for vegetarianism and limited meat consumption in Western countries [36]. The discrepancies in results may derive from varying views on the health benefits of meat. Some people believe that meat consumption is unhealthy [18], but others see meat as healthy [36].

The beliefs of significant others, which can include family members, blogger friends, etc., expressing expectations to eat less meat reflect social influences and can be perceived as a kind of social pressure to behave in a certain way. The motivation to meet these expectations increases the likelihood of demonstrating the expected behavior. However, in our study, a higher score for subjective norms lowered the chances of declaring an intention to reduce meat consumption. This may indicate that significant others expect meat consumption, which is not surprising in a community attached to meat.

## 5. Strength and Limitations

The strength of the study is a representative sample, which provides great potential for scientific and practical use in developing various strategies aimed at more sustainable consumption. The methodology can be applied to different populations, enabling comparisons that consider cultural differences that are important in determining the consumption of meat and plant-based foods. The application of the TPB allows the results of the study to be integrated into broader considerations of consumers’ behavioral motivation to change nutritional habits towards more sustainable consumption.

Nevertheless, the study has some limitations related to the data, which may contribute to some bias. One of them is that the study relied on self-reported information. The study was conducted in the context of one country, and as such, future studies should be expanded to other populations to learn about differences and similarities resulting from sociocultural backgrounds.

Only general concepts, i.e., “less meat” and “more plant food”, were included in the survey. This provides an opportunity to gain insight into general trends in changing consumer behavior, but more specific information on individual foods within each category is missing (i.e., so-called red meat versus white meat). In addition, only the overall concept of “significant others” was considered. The importance of “significant others”, on the other hand, can vary significantly depending on whether they are doctors, nutritionists, teachers, celebrities, friends, etc. In addition, the importance of these people in decision-making can vary considerably, especially across categories by education.

Although there are still some criticisms of the Theory of Planned Behavior (TPB) [51], some elements of the TPB were used in the development of the methodological approach. The partial use of the model is a limitation of the study, but on the other hand, the inclusion of other factors is a response to criticism of the theory that points to the fact that, among other things, all components and pathways in TPB are considered rational without taking into account unconscious factors and their effects on behavior [95]. Although Ajzen argued that background information such as demographics, emotions, personality traits, general values, etc. only affect beliefs and thus indirectly influence intention and behavior [51], in this study, some of these variables (values and habits) were included as directly influencing intention, which is a strength of this study.

## 6. Conclusions

Habitual frequency of eating plant foods, negative feelings about meat, and environmental identity had a stimulating effect on the intention to eat more plant foods and less meat, while experiencing pleasure in eating meat had a limiting effect on the intention to eat more plant foods and less meat. In addition, habitual frequency of meat consumption and subjective norms reduced the likelihood of planning to eat less meat, while no predictive effect was observed for the intention to eat more plant foods. The similarity of the predictive effect of certain factors for eating less meat and more plant-based foods should be considered when developing strategies aimed at more sustainable food consumption. Educational and promotional activities to raise awareness of the link between food consumption and the environment can have a strong impact on eating less meat and more plant-based food, even among those strongly accustomed to meat consumption.

Differences in intentions to eat less meat and more plant foods among men and women, and among people with different levels of education, indicate that especially among men and less educated individuals, changing past behaviors related to the amount of meat and plant foods eaten might prove challenging. Further studies should examine the predictive effect separately in groups identified by age and gender.

## Figures and Tables

**Table 1 nutrients-16-01646-t001:** Intentions to eat more plant food and less meat next year according to sociodemographic characteristics (N = 1003).

Socio-Demographic Characteristics	Total Sample % (N) *	Intentions to Eat			
		More Plant Food (P) **		Less Meat (M) ***	
		Yes % (N)	No % (N)	Yes % (N)	No % (N)
Total sample	100.0 (1003)	60.4 (606)	39.6 (397)	42.1 (422)	57.9 (581)
Gender; *p* < 0.001 (P) **; *p* < 0.001 (M) ***					
Male	48.2 (482)	52.6 (254)	47.4 (229)	35.4 (171)	64.6 (312)
Female	51.8 (520)	67.7 (352)	32.3 (168)	48.3 (251)	51.7 (269)
Age (years); *p* = 0.010 (M)					
18–24	10.4 (104)	57.7 (60)	42.3 (44)	27.9 (29)	72.1 (75)
25–34	19.2 (193)	63.7 (123)	36.3 (70)	42.5 (82)	57.5 (111)
35–44	20.4 (205)	64.4 (132)	35.6 (73)	40.5 (83)	59.5 (122)
45–54	16.2 (162)	56.8 (92)	43.2 (70)	39.5 (64)	60.5 (98)
55–64	22.0 (221)	59.3 (131)	40.7 (90)	49.3 (109)	50.7 (112)
65 and above	11.8 (118)	57.6 (68)	42.4 (50)	46.6 (55)	53.4 (63)
Education; *p* = 0.016 (P)					
Primary	10.0 (100)	54.0 (54)	46.0 (46)	37.0 (37)	63.0 (63)
Vocational	17.9 (180)	52.2 (94)	47.8 (86)	36.7 (66)	63.3 (114)
Secondary	40.2 (403)	61.8 (249)	38.2 (154)	41.2 (166)	58.8 (237)
Higher	31.9 (320)	65.3 (209)	34.7 (111)	47.8 (153)	52.2 (167)
Place of residence					
A village	37.6 (377)	60.5 (228)	39.5 (149)	40.1 (151)	59.9 (226)
A town with less than 20,000 inhabitants	13.9 (139)	63.3 (88)	36.7 (51)	43.2 (60)	56.8 (79)
A city with 20,000–100,000 inhabitants	18.6 (187)	59.9 (112)	40.1 (75)	45.5 (85)	54.5 (102)
A town with 100,001–200,000 inhabitants	10.4 (104)	57.7 (60)	42.3 (44)	38.5 (40)	61.5 (64)
A city with 200,001–500,000 inhabitants	9.3 (93)	57.0 (53)	43.0 (40)	37.6 (35)	62.4 (58)
A city with over 500,000 inhabitants	10.3 (103)	63.1 (65)	36.9 (38)	49.5 (51)	50.5 (52)
Financial situation					
There is enough for everything without much saving	16.9 (170)	62.4 (106)	37.6 (64)	45.9 (78)	54.1 (92)
We live frugally and we have enough means for everything	37.6 (377)	58.6 (221)	41.4 (156)	37.9 (143)	62.1 (234)
We live very frugally to save for major purchases	26.8 (269)	61.3 (165)	38.7 (104)	42.0 (113)	58.0 (156)
There is only enough money for the cheapest food and clothing	8.8 (88)	61.4 (54)	38.6 (34)	50.0 (44)	50.0 (44)
There is only enough money for the cheapest food, not enough for clothing	4.1 (41)	58.5 (24)	41.5 (17)	43.9 (18)	56.1 (23)
There is not enough money even for the cheapest food and clothing	1.3 (13)	76.9 (10)	23.1 (3)	46.2 (6)	53.8 (7)
It’s difficult to say	4.5 (45)	57.8 (26)	42.2 (19)	44.4 (20)	55.6 (25)

* N—number of participants; ** P—significant differences regarding plant food; *** M—significant differences regarding meat; Chi-square test.

**Table 2 nutrients-16-01646-t002:** Habits, subjective norms, attitudes towards meat, and self-identity in the study sample (N = 1003).

Variables	Mean Value	Standard Deviation	Mode	Median	Range
Habit (H): Frequency of eating meat food *	1.26	0.86	1.50	1.14	0–6 *
Habit (H): Frequency of eating plant food *	1.95	1.17	1.06	1.56	0.6–6 *
Subjective norm: Normative beliefs **	3.19	1.04	3	3.00	1–5 **
Subjective norm: Motivation to comply with others **	2.59	1.21	3	3.00	1–5 **
Subjective norm ***	5.79	1.89	6	6.00	2–10 ***
Attitude towards meat (A): Hedonism **	3.58	0.98	4	3.67	1–5 **
Attitude towards meat (A): Negative feelings **	2.32	0.99	1	2.33	1–5 **
Attitude towards meat (A): Beliefs about eating meat **	3.63	0.90	4	3.67	1–5 **
Self-identity (I): a person who cares about health **	3.73	0.94	4	4.0	1–5 **
Self-identity (I): a person who values tradition **	3.67	1.07	4	4.0	1–5 **
Self-identity (I): a person focused on the natural environment **	3.54	0.98	4	4.0	1–5 **

* the sum of the daily frequency of intake of 3 food groups; ** a 5-point scale ranging from 1 (totally disagree/totally not) to 5 (totally agree/very much); *** the sum of two 5-point scales ranging from 1 (totally not) to 5 (very much).

**Table 3 nutrients-16-01646-t003:** Intentions to eat more plant food and less meat with regard to eating habits, subjective norms, attitudes towards meat, and self-identity in the study sample (N = 1003).

Variables	Intentions to Eat					
	More Plant Food			Less Meat		
	Yes *	No *	*p*-Value **	Yes *	No *	*p*-Value
Habit (H): frequency of eating meat food	1.23; 0.86	1.30; 0.86	0.198	1.12; 0.79	1.36; 0.89	<0.001
Habit (H): frequency of eating plant food	2.16; 1.20	1.63; 1.06	<0.001	2.14; 1.22	1.81; 1.12	<0.001
Subjective norms	5.74; 1.83	5.86; 1.96	0.337	5.45; 1.81	6.03; 1.91	<0.001
Attitude towards meat (A): Hedonism	3.31; 0.98	3.99; 0.83	<0.001	3.08; 0.94	3.95; 0.84	<0.001
Attitude towards meat (A): Negative feelings	2.58; 0.98	1.91; 0.85	<0.001	2.80; 0.92	1.97; 0.88	<0.001
Attitude towards meat (A): Beliefs about eating meat	3.43; 0.86	3.94; 0.86	<0.001	3.28; 0.86	3.88; 0.84	<0.001
Self-identity (I): a person who cares about health	3.81; 0.91	3.60; 0.97	<0.001	3.77; 0.95	3.69; 0.94	0.178
Self-identity (I): a person who values tradition	3.63; 1.08	3.73; 1.04	0.143	3.54; 1.09	3.76; 1.04	0.002
Self-identity (I): a person focused on the natural environment	3.72; 0.92	3.27; 1.02	<0.001	3.73; 0.91	3.41; 1.02	<0.001

* mean values and standard deviations in the subgroups identified according to declared intentions to eat more plant food and less meat; ** significance, Student’s *t*-test (t).

**Table 4 nutrients-16-01646-t004:** Associations between frequency of eating and intentions to eat more plant food and less meat, eating habits, subjective norms, attitudes, and self-identity in the study sample (N = 1003).

Variables	Frequency of Eating				Intentions to Eat			
	Plant Food		Meat		More Plant Food		Less Meat	
	r **	*p*-Value	r	*p*-Value	r	*p*-Value	r	*p*-Value
Frequency of eating meat food (H) *	0.272	<0.001			−0.041	0.198	−0.136	<0.001
Frequency of eating plant food (H) *			0.272	<0.001	0.221	<0.001	0.140	<0.001
Subjective norms	−0.036	0.252	0.166	<0.001	−0.030	0.337	−0.150	<0.001
Hedonism (A) *	−0.078	0.014	0.225	<0.001	−0.338	<0.001	−0.438	<0.001
Negative feelings (A) *	0.036	0.255	−0.090	0.004	0.327	<0.001	0.415	<0.001
Beliefs about eating meat (A) *	−0.028	0.383	0.186	<0.001	−0.277	<0.001	−0.332	<0.001
A person who cares about health (I) *	0.227	<0.001	0.009	0.787	0.106	<0.001	0.043	0.178
A person who values tradition (I) *	0.111	<0.001	0.114	<0.001	−0.046	0.143	−0.099	0.002
A person focused on the natural environment (I) *	0.184	<0.001	−0.024	0.452	0.224	<0.001	0.158	<0.001

* H—habit; A—attitude towards meat; I—identity; ** r—Pearson correlation coefficient.

**Table 5 nutrients-16-01646-t005:** Odds ratios for the intention to eat more plant food and less meat and meat products in the following year.

Variables	Intention to Eat More Plant Food Next Year (Model 1)					Intention to Eat Less Meat and Meat Products Next Year (Model 2)				
	β	e^β^	95CI **		*p*-Value **	β	e^β^	95CI		*p*-Value
Frequency of eating meat food (H) *	−0.065	0.94	0.78	1.13	0.497	−0.225	0.80	0.65	0.98	0.028
Frequency of eating plant food (H) *	0.411	1.51	1.31	1.74	<0.001	0.271	1.31	1.14	1.51	<0.001
Subjective norms	0.029	1.03	0.95	1.11	0.472	−0.138	0.87	0.80	0.95	0.002
Hedonism (A) *	−0.524	0.59	0.47	0.75	<0.001	−0.778	0.46	0.36	0.58	<0.001
Negative feelings (A) *	0.472	1.60	1.35	1.91	<0.001	0.727	2.07	1.74	2.46	<0.001
Beliefs about eating meat (A) *	−0.186	0.83	0.65	1.06	0.141	−0.025	0.98	0.76	1.25	0.844
A person who cares about health (I) *	0.118	1.13	0.94	1.35	0.196	0.038	1.04	0.86	1.26	0.697
A person who values tradition (I) *	−0.088	0.92	0.78	1.08	0.282	−0.067	0.94	0.79	1.11	0.432
A person focused on the natural environment (I) *	0.402	1.49	1.26	1.78	<0.001	0.245	1.28	1.06	1.54	0.009

* H—habit; A—attitude towards meat; I—identity; ** OR—point estimate (e^β^), 95% confidence intervals; significance level of the Wald’s test.

## Data Availability

The data is not publicly available because the data have not yet been made available in ‘publicly available databases’. However, the data presented in the study are available on request from the corresponding author.

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
