# Peer review of "Predictors of Eating Less Meat and More Plant-Based Food in the Polish Sample"

_nutrients, 2024, doi:10.3390/nu16111646_

Round 1

Reviewer 1 Report

Comments and Suggestions for Authors

Sentence on lines 39 and 40 is too vague and it’s meaning is not quite clear.

Paragraph starting at line 344 is way too long. Identify the main themes (age, education), and separate into 2-3 paragraphs

This is a very good paper. Thank you for doing this research and writing this article. The English is excellent and the ideas are understandable and interesting.

However, the discussion and conclusion are too long. The results are repeated multiple times at various points in the paper. The discussion often repeats the finding and then highlights the context for a result, a particular relationship (eating less meat and subjective norms, for example) and then discusses how this should inform policy/research/messaging going forward. Then the same thing happens with the next relationship discussed (eating less meat and frequency of meat consumption, for example). This makes the whole section very long and quite redundant. Perhaps there could be a separate section that addresses how these results inform future policy/messaging/research, rather than putting it in throughout the discussion/conclusion. Then the results and the context for the results can be streamlined to some degree, without repeating so many of the results over again in the conclusion.

Author Response

We would like to kindly thank the Reviewer for the time and effort taken to read and review our article. We greatly appreciate all the comments and suggestions.

Please find included the all responses.

Kind regards

Reviewer 2 Report

Comments and Suggestions for Authors

The issue of the determinants of the intention to eat more plant-based diets and less meat is of scientific interest. This cross-sectional study analyses the attitudes towards eating meat, habitual meat eating, and subjective norms and self-identity of n=1003 Polish adults. Results show predictors of the intention to eat less meat, more plant-based food or both.

I have a number of questions:

Introduction

Please revise the text Lines 53-63, the use of the terms “plant-based food”, “plant-based diet”, and “plant food”. I agree that it may be that following a plant-based diet (vegetarian, vegan) “is seen as inconvenient because of the difficulty of preparing meals, as it requires a lot of time and skill, expensive ingredients, etc.” but at the same time preparing a specific plant-based food (e.g. a veggie burger) may be very easy and quick. Regarding the sentence “Moreover, plant food is less preferred than food of animal origin due to certain sensory characteristics, including taste [21–24]”, it is known that animal foods are highly palatable, but this does not mean that plant food is always less preferred. I find that these expressions are too blunt. I suggest, adding words, such as “ in several surveys”, “different studies report”, or similar.

Methods

2.1. Lines 135-137. The inclusion and exclusion criteria are not detailed. Please indicate the age range, whether patients with disease were included or not, whether vegetarian or vegan were included or excluded, etc. In addition, information on the method of recruitment will be useful, e.g. by e-mail, posters, social media, etc. Also indicate the location or region where the sample was taken.

It seems that the study was designed to have a sample size of ~ 1000. Is there any basis for this number?. Usually, a sample size calculation is performed based on representativeness (the sample is representative of the population) or a quantitative variable that is considered as the principal variable.

2.2. I miss the list of the 6 food groups. I suggest deleting the text between the brackets, and giving the 6 groups, with examples in brackets if needed. In my opinion, there are plant-based foods difficult to classify into one group. Have you had such as difficulties?

2.3. It is not clear what is the reference group, that is defined as “namely the significant others”. It will be easier if the questions are given in a supplementary table (see below).

Line 170-170. The 7 statements by ref. 37 and the 2 statements by ref. 59 should be mentioned. If possible, list all the questions. This can be presented in a supplementary file and will be very useful for the general reader and to facilitate replication of the study.

2.4. Statistics. Lines 212-213. Chi-square test and Student´s t-test were used to determine differences, between variables or between subgroups? Please revise the sentence.

Results

Table 2. The variables are likely to be non-normally distributed. In this case, it is preferable to present the data as median and range rather that mean and SD. In addition, the foot of this table is not clear. Please revise.

Table 3. Indicate what the numbers are, e.g. 1.23; 0.81 in the first cell.

Table 5. Too many *, **, ***, to represent different things. You can just indicate: H, habit; A, attitude; 95CI, 95% confidence interval; etc….

Discussion

There may be a bias in this study. For example, if a consumer follows a vegan diet, her/his answers may be in the direction of no change (no intention to eat more plant food, no intention to eat less meat). This could be included in the discussion.

The strengths and limitations should be clearly separated.

The conclusions are very long. Try to shorten this section and clarify what the main finding are.

Minor points

Please revise the pagination as all pages are 5 of 21.

Comments on the Quality of English Language

English language is adequate, minor revision is required

Author Response

Dear Reviewer,

We would like to thank for the all suggestions you have indicated. Please find enclosed the responses to your suggestions.

Kind regards

Round 2

Reviewer 2 Report

Comments and Suggestions for Authors

The authors have revised and corrected the manuscript. They have answered all my questions, and the article has been improved. I think has enough quality to be accepted for publication.

Minor editing is still needed. e.g. Table 5 the use of * in the first column inside or outside parenthesis and it should be removed from eβ*

Comments on the Quality of English Language

Quality of the English language is good.

Author Response

Dear Reviewer,

Thank you for your suggestions. Please find the corrected paper. I hope we have understood well your suggestions.

Kind regards,